# StationRank: Aggregate dynamics of the Swiss railway

**Georg Anagnostopoulos**©*, **Vahid Moosavi**

Department of Architecture, ETH Zurich, Zurich, Switzerland

* mail@anagno.com

## Abstract

Increasing availability and quality of actual, as opposed to scheduled, open transport data offers new possibilities for capturing the spatiotemporal dynamics of railway and other networks of social infrastructure. One way to describe such complex phenomena is in terms of stochastic processes. At its core, a stochastic model is domain-agnostic and algorithms discussed here have been successfully used in other applications, including Google's PageRank citation ranking. Our key assumption is that train routes constitute meaningful sequences analogous to sentences of literary text. A corpus of routes is thus susceptible to the same analytic tool-set as a corpus of sentences. With our experiment in Switzerland, we introduce a method for building Markov Chains from aggregated daily streams of railway traffic data. The stationary distributions under normal and perturbed conditions are used to define systemic risk measures with non-evident, valuable information about railway infrastructure.

## Introduction

The present study provides a stochastic framework for studying interactions between localities (stations) that are connected via railway infrastructure. We construct rankings of stations by means of aggregating daily train flows. In terms of scope, we cover the whole Swiss railway network over a period of one month. Our hypothesis is that we can obtain a quite detailed idea of the spatiotemporal dynamics involved just by utilizing a single openly accessible data set, at low computational cost and without assumptions involving substantial domain-specific knowledge.

Existing parametric tools, such as agent-based models [1], have been applied to many engineering problems related to complex systems, including analysis of public transportation networks (PTN). Nevertheless, their success relies on the acceptance of a predefined set of properties [2], whose gradual increase in pursuit of realism is bound by the curse of dimensionality [3]. On the other hand, purely stochastic models implicitly capture complexity in a probabilistic manner. In PageRank [4] for example, websites are indirectly represented by their relations to other websites without relying on predefined semantics. Since then, there have been multiple adaptations of this idea, best summarized in *"PageRank beyond the web"* [5].

**Data Availability Statement:** All relevant data and code used in this work are accessible from: https://github.com/GAnagno/Social-Web/blob/master/Stationrank.ipynb.

**Funding:** The author(s) received no specific funding for this work.

**Competing interests:** The authors have declared that no competing interests exist.

Topic-specific publications about the spatiotemporal dynamics of railway systems are scarce as literature is dominated by studies focusing on road traffic. For example an early paper [6] on a *"graph-theoretical analysis of the Swiss road and railway networks over time"*, actually only covers the Swiss freeways, not railways. Later, a methodology for assessing the structural and operational robustness of railway networks with examples from Switzerland was developed, but it was stationary and acknowledged that it would take *"much effort for turning the stationary model into a dynamic one"* [7]. Nevertheless, this is the closest alternative source known to the authors in terms of comparing results. A recent study [8] took an aggregated approach in order to statistically explain the spatiotemporal ramifications of an actual, extremely rare disruptive event, but their methodology has a rather forensic character. There are also other relevant frameworks [9], which unfortunately exclude railways from consideration.

Our approach builds on the foundations laid out by [10] on road network dynamics and adapts it for a real railway network by introducing a novel model architecture. StationRank preserves actual train stations by avoiding aggregation in space and is thus more detailed than earlier [11] or current [9] approaches that employ spatial binning. Here we also complement previous research [12] on systemic risk [13] with spatial aspects.

## The method

We study the dynamics of the Swiss railway network over a period of one month via characteristics of Markov Chains (MC) [14]. Originally conceived for statistical analysis of texts, MC have proven useful in various fields such as search engines [4], traffic dynamics [9–11, 15] and econometrics [12, 13]. More formally, MC are defined as stochastic processes that display the *Markov Property*

$$p(x_{k+1} = S_{k+1} | x_k = S_k, x_{k-1} = S_{k-1}, \ldots, x_0 = S_0) = p(x_{k+1} = S_{k+1} | x_k = S_k), \qquad (1)$$

which means that the probability of a random variable $x$ being at state $S_{k+1}$ at time step $k + 1$ only depends on its state $S_k$ at time step $k$ [10]. We can think of $S$ as the position of a particle $x$ following a random walk. It is worth remarking that *"vehicles do not actually follow a random walk; however, by analogy with similar approaches like the PageRank model . . ., the underlying stochastic process is viewed as a mechanism for obtaining useful information, rather than a literal representation of the behaviour of a single vehicle"* [10].

Overall, we have a day-to-day dynamic process, where each day is one discrete-time, finite-state homogeneous MC [10, 11, 16] for itself. In a way, we have two levels of aggregation and discretization [17]: days within a month and minutes within a day. Each MC can be stored in the form of a row-stochastic, non-negative $n \times n$ *transition probability matrix* $\mathbb{P}$, where $n$ is the number of states and $\mathbb{P}_{ij}$ is the transition probability from state $S_i$ to state $S_j$ [10].

The majority of existing stochastic traffic models focus on road networks and have established methods for calculating the transition matrix. For example [10, 16] consider turning probabilities at road intersections and mean travel times at road segments in order to construct the matrix, while [11] samples GPS traces of taxicabs with a rectangular grid and constructs the Markov matrix based on grid cell transition times. Other approaches, as in [9], solve traffic assignment optimization problems based on synthetic behavioral data and domain-specific assumptions which result in origin-destination matrices.

Our methodology for computing the transition probability matrices is more tailored towards utilization of actual public transport itineraries (trips). Trips are stop sequences with continuous departure and/or arrival times at discrete locations (stops). Every trip consists of dwell and running times [15]. After discretization [17], dwell times are assigned to actual

stations (dwell states) and running times are assigned to virtual transit stations (running states). We can think of these stations (actual or virtual) as Markovian states $S$. By simply counting the total number of transitions from one state $S_i$ to another state $S_j$, we obtain a weight matrix $\mathbb{A}$. Off-diagonal entries $\mathbb{A}_{ij}$ represent change of state and diagonal entries $\mathbb{A}_{ii}$ represent preservation of state. We can show that it is pretty straightforward to obtain the matrix $\mathbb{P}$ given $\mathbb{A}$.

Regarding the aggregation, we group stops by trip id and day of operation. A day of operation is not the same as a calendar day. Consider the following: a single trip around midnight might span two calendar days, but occurs in the same day of operation. Then, for each day of operation, we convert all trips to special continuous trajectory objects [18] which we discretize: https://bit.ly/3iKLx7A. The resulting discrete trajectories are in fact Markovian state sequences.

**Example 1**. **Markovian state sequence**. $(A)$, $(A \rightarrow B)$, $(A \rightarrow B)$, $(B)$, $(B)$, $(B)$, $(B \rightarrow C)$, . . ., $(X \rightarrow Y)$, $(Y)$, $(Y)$, $(Y \rightarrow Z)$, $(Z)$.

Letters denote actual train stations. By computing the daily total number of observed transitions between a state $S_i = (A)$ and a state $S_j = (A \rightarrow B)$, we get $\mathbb{A}_{ij}$. The weight matrix $\mathbb{A}$ is actually the adjacency matrix of a directed multi-graph $\mathcal{D}$. From $\mathcal{D}$, we extract the strongly connected component $\mathcal{G}$, using algorithm [19]. Finally, we compute the row-stochastic transition probability matrix $\mathbb{P}$ from the adjacency matrix of $\mathcal{G}$ by ensuring that every row sums up to 1.

Regarding the discretization, we assume a time step of 1min. At every time step, the random variable should be unambiguously assigned to just a single state. It does not matter if $S_k$ is an actual train station (dwell state) or a running state between stations, as long as the function $k \mapsto S_k$ is injective. Now, if we consider 10min discretization, a suburban train with frequent stops might have traversed multiple states within this time step, which leads to a contradiction (state ambiguity). For other systems, such as bus or tram networks, one might even want to consider a discretization of 30sec. Of course there exist other valid Markovian frameworks with 10min time step [11], but they employ spatial binning which is not meaningful in our setup.

## The data

The actual data [20] is published by the Swiss Federal Railways (SBB) on a daily basis. In contrast to timetables, each day of actual data is a unique collection of routes that enables us to study the real behavior of the network. The data set is multi-modal, including trains, buses, trams, metro and boats. In the present work we take only train data into consideration under the assumption that the corresponding networks are of different nature: trains do not typically share stations with boats and have also very different speed. The data set is also multi-jurisdictional, involving many different operators, a challenging setup from the perspective of homogeneity and data integrity. It is important to note that such data does not actually represent passengers or goods flow on the network, only trains. Nevertheless, even at this detail level, we show that it is possible to gain interesting insights about the system. Generally, the actual data is a valuable source also for other investigations such as spatiotemporal prediction problems.

## The model

We propose here a hybrid model which combines features of both **primal** and **dual** networks [10, 16, 21–23]. A primal approach would represent stations as nodes and their connections as edges. The inverse, dual approach would turn the connections into nodes and the stations into edges. The advantages or drawbacks of primal and dual graphs have already been thoroughly

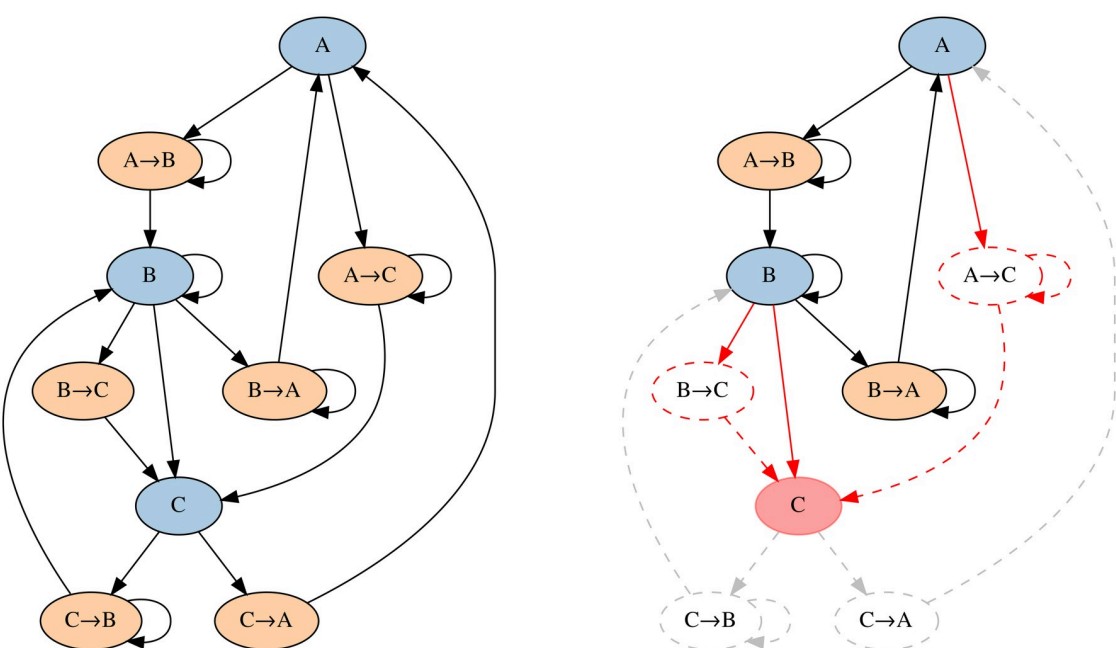

**Fig 1. StationRank architecture.** Example sub-network incorporating temporal and modal aspects. The nodes are Markovian states. We distinguish between dwell states which are actual stations, such as (*A*), and running states which are transits between stations, such as (*A* → *B*). Stations like (*B*) and (*C*) might be both directly and indirectly linked due to differences in speed of connecting trains. A disruption at node (*C*) can be caused by means of reducing all of its inflows.

discussed in other works [22, 23], but, to our best of knowledge, merging the two approaches offers an alternative perspective that has not enjoyed enough attention yet. Before introducing a new network architecture, it is important to explain the concept of space.

In the context of PTN, a space is a way to define network topology. Some examples of spaces are *L*-space, *P*-space, *B*-space and *C*-space [24]. Each station in *L*-space is represented by a node. Links between nodes indicate that the respective stations are consecutively served by at least one route [24, 25]. If stations are connected only when they are consecutive stops of at least one vehicle, then we have a **space-of-stops** as a special case of *L*-space. If stations are connected only when there is a physical direct link (such as rail tracks) with no other station in between, then *L*-space is actually a **space-of-stations** [21]. Space-of-stops coincide with space-of-stations when vehicles stop at every station. An extended *L′*-space allows weighted multi-links.

In *P*-space any two stations, consecutive or not, that share at least one common route are linked. This representation is also called **space-of-changes** [21] because each route results in a clique including all stations that can be reached without changing vehicle. *B*-space is a bipartite graph where stations are linked to serving routes. Finally, *C*-space does not include any stations at all, only routes. In *C*-space connected routes share at least one station.

Our network is defined in *L′*-space. We introduce a generic **space-of-states**, where a state can be a stop or a transit, Fig 1. In total, we consider 1660 stops. StationRank incorporates temporal as well as modal aspects. Dwell states (actual stations) might be directly or indirectly connected via running states (transits). An example sub-network in the canton of Grisons can be found in the supporting information section, see S1 Fig. Space-of-states explicitly captures temporal dynamics as it does not assume any fixed topology (e.g. roads or rail-tracks) and can

easily support construction of discrete-time, finite-state homogeneous MC without spatial sampling. This novel approach is easily applicable also to other systems, such as air traffic.

## Building ergodic MC

For every day, we assume a *strongly connected* directed graph $\mathcal{G}$ which also implies that the corresponding matrix $\mathbb{P}$ should be *irreducible* [10, 11]. All states of an irreducible and aperiodic MC, commonly referred to as *ergodic*, are **positive recurrent** [26] with

$$\mathbb{E}(T_i|x_0 = S_i) < \infty, \tag{2}$$

where $T_i$ is the time of first return to $S_i$. This means that a train is expected to return to an initial state within a finite number of steps.

Matrices built from actual data do not always contain only positive recurrent states. There are two kinds of states which need special attention: **null-recurrent** [11, 26] and **absorbing states** [4, 12, 27].

- State $S_i$ is null-recurrent if $\mathbb{E}(T_i|x_0 = S_i) = \infty$

- State $S_i$ is absorbing if $p(x_{k+1} = S_i|x_k = S_i) = 1$

Essentially, returning to a null-recurrent or escaping from an absorbing state is impossible. There might be whole groups of states that behave as either absorbing or null-recurrent. A description of all possible network artifacts, including in-components, out-components, in-tendrils, out-tendrils, tubes and disconnected components was first reported in [28]. One way to overcome these issues is random teleportation [4, 11].

Random teleportation blurs out the originally sparse transition probability matrix with a homogeneous dense matrix. This is a relatively high price to pay in order to achieve ergodicity as it significantly alters network topology and results in unnecessarily long computation times. There are also other problems such as exploding mean first passage times and unacceptable values for the Kemeny constant. We introduce these concepts in the next section. The cleanest solution in mathematical terms is altogether refraining from teleportation [16]. While random teleportation works for web pages, it is less meaningful in the context of transportation systems, so we limited our analysis to the strongly connected component which can be found by fast depth-first search [19]. Because the number of null-recurrent and absorbing stations compared to our network size is in the vicinity of 7.5%, we still have a strongly connected component near 92.5% of the original network which is an acceptable trade-off, see also S2 Fig.

## Eigenspectra and the Kemeny constant

For every ergodic MC, according to the Perron-Frobenius theorem [10, 11], all eigenvalues of $\mathbb{P}$ are within a *spectral radius* of 1 with the largest of them, called the *Perron root*, being always equal to 1 and unique. The corresponding left-hand Perron eigenvector is also unique and defined by

$$\pi^T \mathbb{P} = \pi^T, \tag{3}$$

such that $\pi > 0$ and $\|\pi\|_1 = 1$. Each entry $\pi_i$ of the *stationary distribution vector* $\pi^T$ represents the long-run time fraction of being in the respective state [10]. The term stationary in the context of Markov Chains is not referring to a stationary system. See [7] for an example of a stationary model. Here the stationary distribution $\pi$ represents occupancy of stations at a theoretical state of equilibrium, it shows a trend. Generally, $\pi$ is just one property of the underlying Markovian system as described by the transition probability matrix $\mathbb{P}$. The Markov

matrix $\mathbb{P}$ can support much more demanding analysis such as calculation of mean first passage times, network resistance estimation, link recommendation, etc. Of particular interest is also the second eigenvector of $\mathbb{P}$. It has been shown [10, 29] that if the respective eigenvalue is real, the second eigenvector associates nodes to weakly connected sub-communities.

By solving $\pi^T \mathbb{P} = \pi^T$, we obtain the stationary distribution vector $\pi^T$ and also the eigenvalues $\lambda_1 = 1, \lambda_2, \ldots, \lambda_n$. Then, we can compute the *Kemeny constant K* which is an intrinsic quantity of every MC [10–12, 16].

$$K = \sum_{j=2}^{n} \frac{1}{1 - \lambda_j} \tag{4}$$

This spectral formulation is very straightforward to calculate, but difficult to interpret. On the other hand, a formulation in terms of *mean first passage times* $m_{ij}$ grants us more insight to the inner workings of $K$. Mean first passage time is the expected number of steps from an origin state $S_i$ to a destination state $S_j$.

$$K = \sum_{j=1}^{n} m_{ij} \pi_j \tag{5}$$

Eq (5) is yet another, more intuitive way to calculate Kemeny's constant as the average mean first passage time from any origin state to any other destination state chosen according to the probability $\pi_j$ and shows that $K$ is independent from the choice of the initial state $S_i$. $K$ is an excellent indicator of network efficiency and can be used for comparing different systems. The constant should be normalized by the duration of aggregation in order to estimate the average travel time in minutes.

Obviously now, in addition to $\pi$ we also need to calculate mean first passage time, which in turn requires the calculation of the group (special case of Drazin) inverse $\mathbb{Q}^\#$, where $\mathbb{Q} = (\mathbb{I} - \mathbb{P})$ and $\mathbb{I}$ is the $n \times n$ identity matrix. According to [10, 30], $\mathbb{Q}^\#$ is the group inverse of $\mathbb{Q}$, if and only if:

1. $(\mathbb{Q})(\mathbb{Q}^\#)(\mathbb{Q}) = (\mathbb{Q})$

2. $(\mathbb{Q}^\#)(\mathbb{Q})(\mathbb{Q}^\#) = (\mathbb{Q}^\#)$

3. $(\mathbb{Q})(\mathbb{Q}^\#) = (\mathbb{Q}^\#)(\mathbb{Q})$

The mean first passage time can then be calculated with Eq (6). As we will see in the results section, mean first passage time provides a more intuitive understanding of $K$ and has a quite interesting distribution and interpretation in space.

$$m_{ij} = \frac{q^\#_{jj} - q^\#_{ij}}{\pi_j} \tag{6}$$

## Sensitivity analysis of transition probability matrices

It has been shown in previous works [10, 12] that, by perturbing the values of the corresponding transition matrix, one can analyze the effect of a disruption at some node on the other nodes and the network as a whole. The original transition matrix $\mathbb{P}$ is thus perturbed into $\tilde{\mathbb{P}} = \mathbb{P} + E$ [10], where each row of the matrix $E$ sums to zero. For a reduction of just one

entry $\mathbb{P}_{ip}$ of the Markov matrix by a quantity $t(\mathbb{P}_{ip})$,

$$E = t \frac{\mathbb{P}_{ip}}{1 - \mathbb{P}_{ip}} e_i [e_i^T \mathbb{P} - e_p^T], \qquad (7)$$

where $e_k$ is a vector of zeroes whose $k^{th}$ entry equals 1 [10] and $\mathbb{P}_{ip} < 1$. If $\mathbb{P}_{ip} = 1$, then $E_{ip} = -t$, $E_{ii} = t$ and all other entries of $E$ are set to zero. Eq (7) reduces the capacity of a single link between any states $S_i$ and $S_p$. Reducing the activity of a node [12], as opposed to a single link, requires a general perturbation strategy. Our strategy is to cause node disruption by weakening certain incoming links, Fig 1. Node disruption involves one or more of the following cases:

- Eq (7) applies to an inflow point $S_i$ directly connected to a target station $S_p$.

- Eq (7) applies to an inflow point $S_i$ indirectly connected via a running state $S_p$.

- Eq (8) applies to an inflow point $S_i$ connected to a target station $S_p$ via a set of states $\mathbb{S}$, where $S_p \in \mathbb{S}$. Eq (8) is our generalization of Eq (7):

$$E = t \frac{\sum_{S_j \in \mathbb{S}} \mathbb{P}_{ij}}{1 - \sum_{S_j \in \mathbb{S}} \mathbb{P}_{ij}} e_i [e_i^T \mathbb{P} - e_{\mathbb{S}}^T], \qquad (8)$$

where every $j^{th}$ entry of $e_{\mathbb{S}}$ equals $\dfrac{\mathbb{P}_{ij}}{\sum\limits_{S_j \in \mathbb{S}} \mathbb{P}_{ij}}$ and $\sum\limits_{S_j \in \mathbb{S}} \mathbb{P}_{ij} < 1$.

We assume a homogeneous reduction of 95% on all inflows. This is a relative measure; applied to a busy hub will emphasize the impact and applied to a remote station will deemphasize the impact. From the resulting perturbed transition probability matrix $\tilde{\mathbb{P}}$, a new stationary distribution $\tilde{\pi}$ can be calculated. Dedicated systemic risk measures can be articulated by comparing the stationary distributions under normal and perturbed conditions.

## Results

Here we highlight a few different use cases that we found interesting for the reader. Even in a highly efficient, optimized and well studied system such as the Swiss railway, there are surprising results when dealing with the daily variation of the actual flows.

### Mean first passage time

Mean first passage time $m_{ij}$ is by definition expected to have very similar probability distribution for different initial stations $S_i$. If, instead, we consider its transpose $m_{ji}$, then we come up with a new kind of accessibility analysis in terms of *remoteness* of train stations, Fig 2. This can be easily verified on a map, see S3 Fig.

### First eigenvector

The first eigenvector of the transition probability matrix $\mathbb{P}$ is equivalent to the stationary distribution of the system. The stationary distribution can also be viewed as a kind of centrality measure, *eigencentrality*, Fig 3. Here the terminology might seem counter-intuitive, but stationarity is actually just another property of a compact *dynamical system representation* [31]. Eigencentrality provides a high level overview of the system and remains largely stable with time, S8 Fig. Nevertheless, closer inspection in terms of descriptive statistics reveals certain

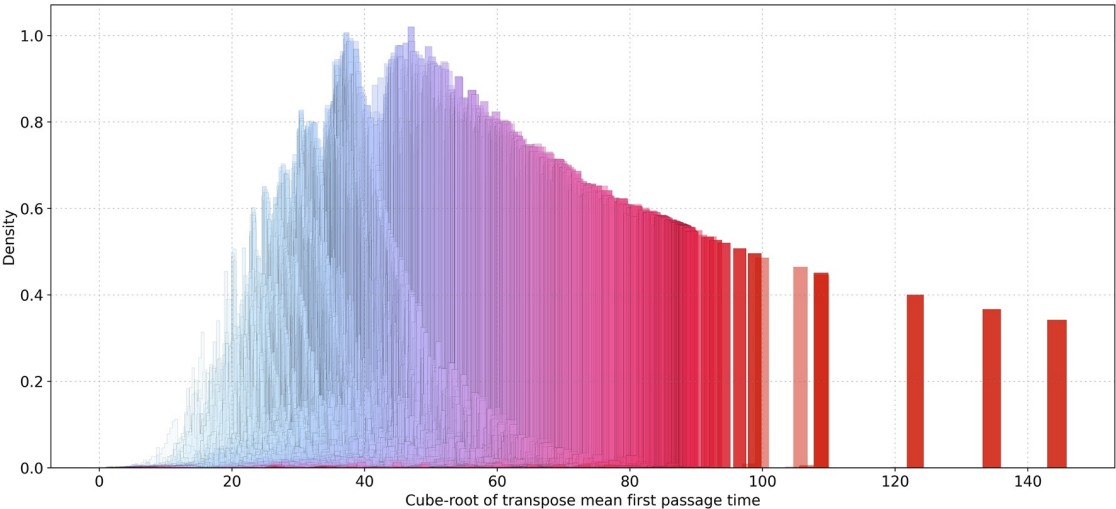

**Fig 2. Probability distribution of transpose mean first passage time.**

trends, as we will see later. For more details and variation, we also need to consult the second eigenvector.

## Second eigenvector

Due to the orthogonality of first and second eigenvector, there is zero correlation between them and thus both convey very valuable and non-overlapping information. Whereas the first one is equivalent to the stationary distribution, the second one is related to the second eigenvalue which measures the rate of convergence to the stationary distribution [10]. As we

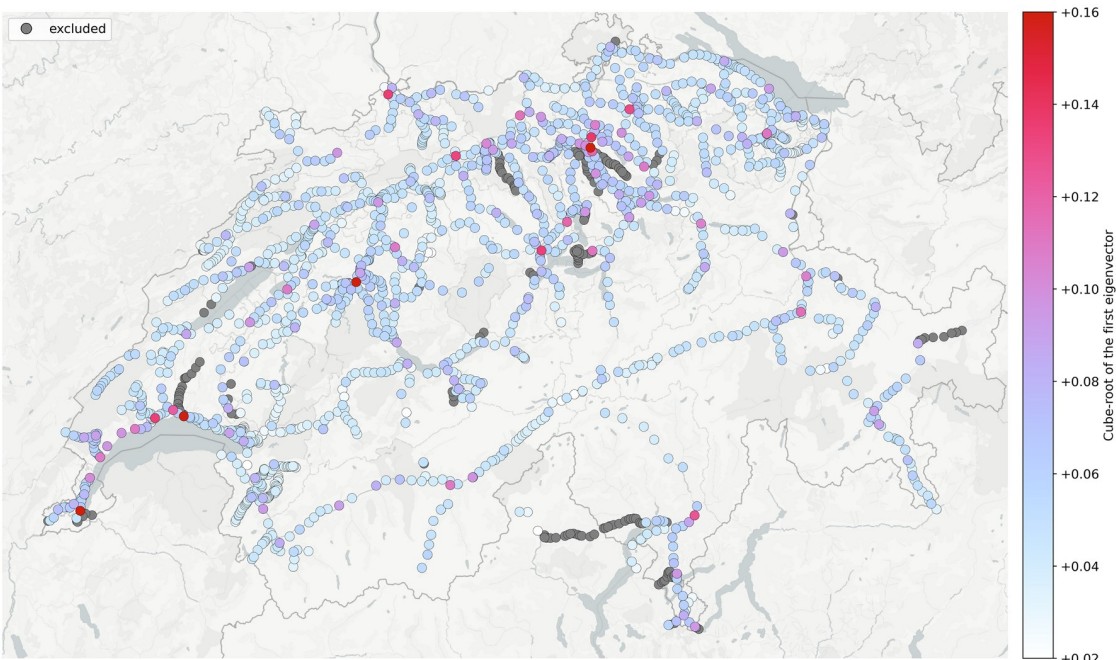

**Fig 3. Single day view of the stationary distribution (1st eigenvector).**

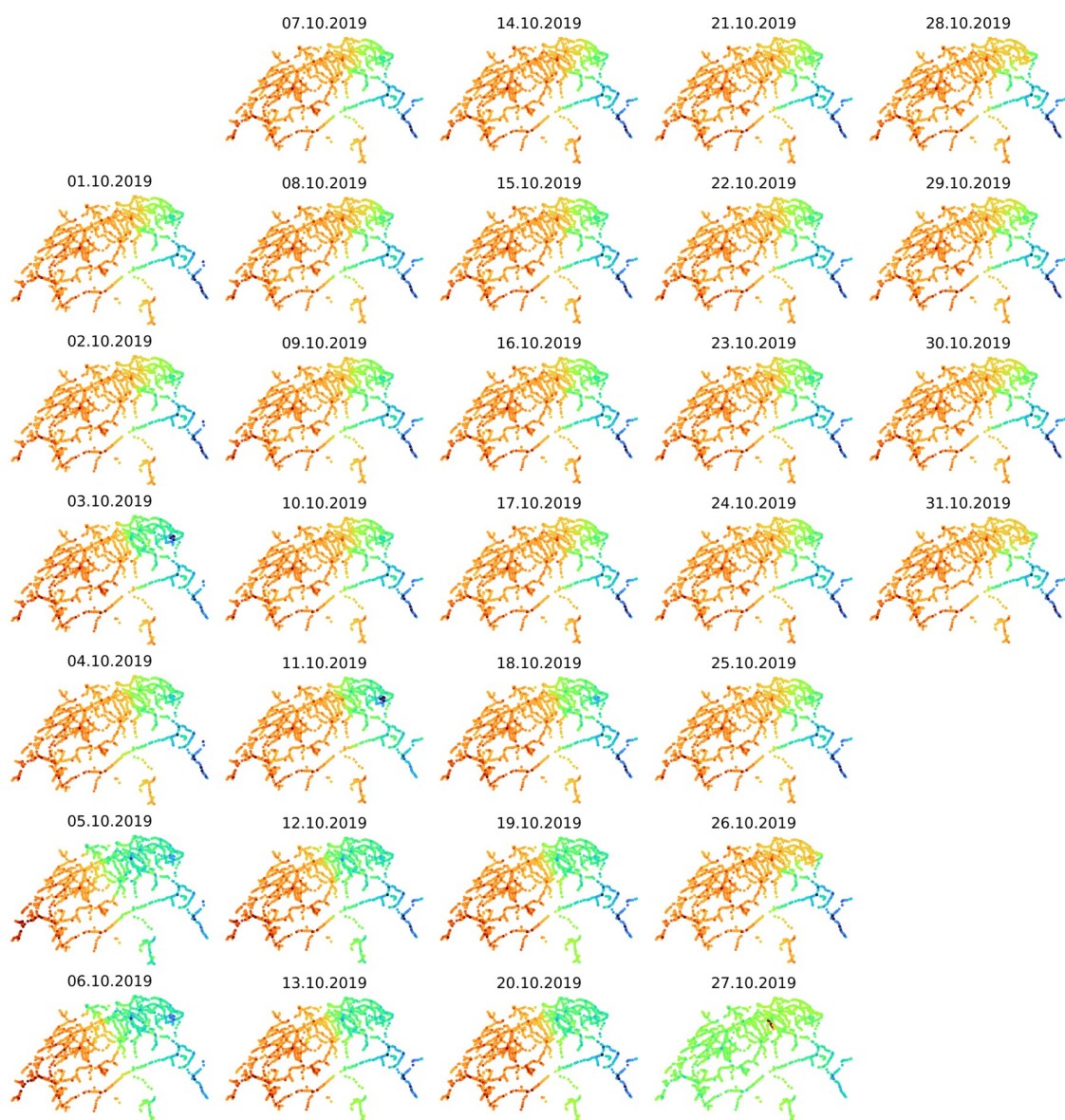

**Fig 4. Spectral clustering in space and time (2nd eigenvector).** Spatiotemporal dynamics of the the Swiss railway's weakly connected sub-communities. There is a stark, gradually attenuating shift in the spectral clusters during weekends.

mentioned before, the second eigenvector is known to be a good indicator for the existence of weakly connected sub-communities and is closely related to the concept of spectral clustering and minimum balanced cuts [29].

Our results are in agreement with the theoretical foundation [10], according to which the eigenvector associated with the second eigenvalue of a Markov chain can be used to detect nearly disconnected groups of states. This is true for the southeastern canton of Grisons and occasionally for the Appenzell district (03.10.2019 & 11.10.2019), Fig 4. Weakly connected routes might be also attributed to construction works (27.10.2019). Surprisingly, the weakly connected sub-communities are not static. A clear shift in the spectral clustering occurs in the first weekend of October (05.10.2019 & 06.10.2019). This could be related to the beginning of the Swiss autumn holidays, as we discuss in the section about dynamics of the system in time.

## Perturbation analysis

By comparing the respective stationary distributions, before and after disruption of a certain station, one obtains a very detailed picture of the dynamics involved. We show the country-wide effect on $\pi$ by a disruption at St. Gallen as an example, Fig 5. Further examples are available on our web app: https://bit.ly/2SJonUB. The reader is also encouraged to experiment with the open source code: https://bit.ly/3fGvCXC.

Generally, positive effect suggests traffic above normal capacity and high congestion as the probability of staying at certain stations rises. Negative effect means reduced traffic as the respective stations operate below normal capacity, but it also means reduced accessibility. See also single day view, S5 Fig. If we compare Fig 5 with Fig 4, then following observation holds:

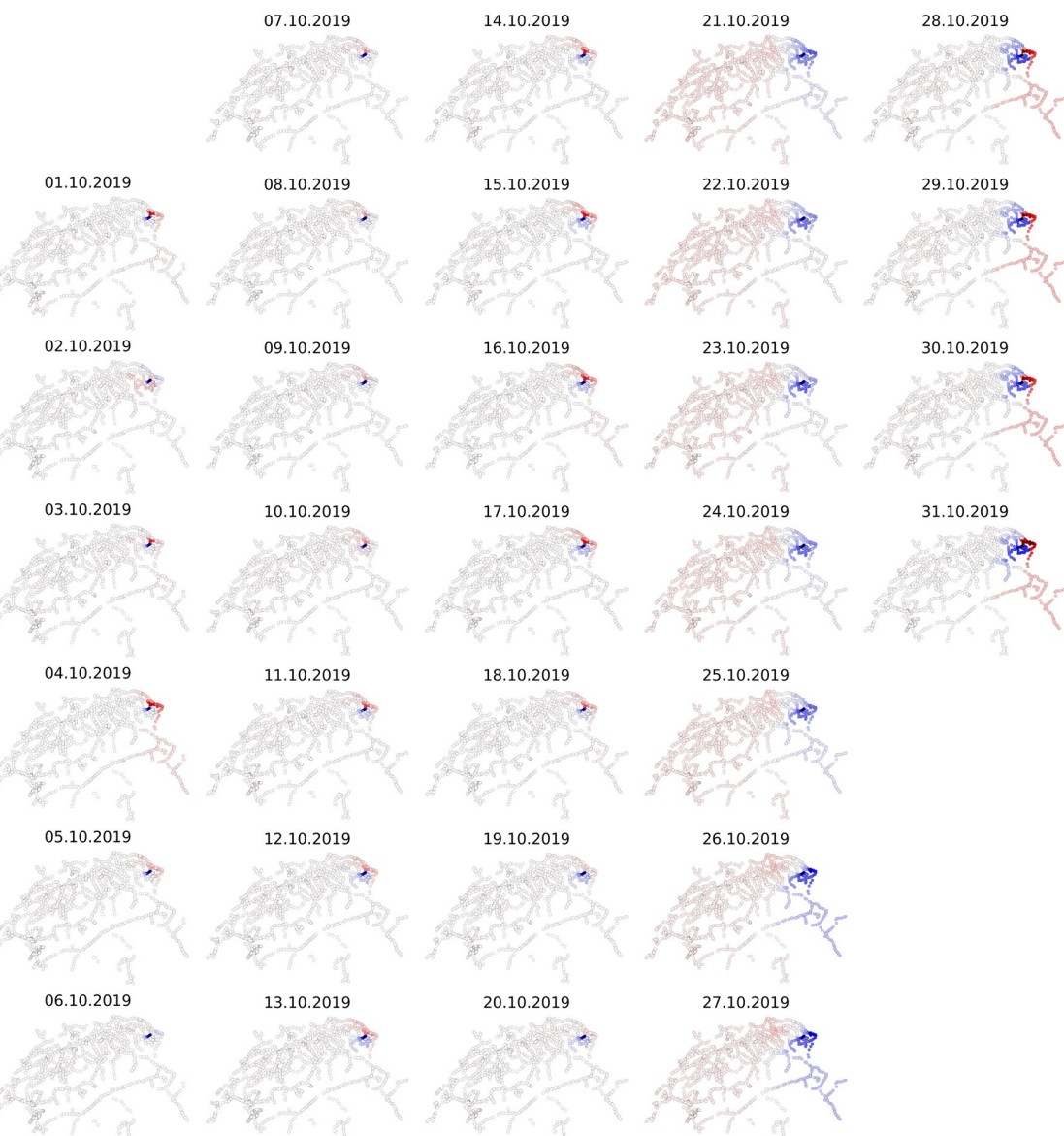

**Fig 5. Countrywide effect on $\pi$ by a disruption at St. Gallen.** Percentages of change with respect to the corresponding stationary distribution. Positive changes (red) suggest increased traffic, negative changes (blue) reduced traffic.

the disruptive effect is minimal when St. Gallen is part of the minor sub-component and gets maximized when the station shifts to the major sub-component.

## Systemic risk measures

Known network measures, such as various centrality measures [6, 7, 32], are purely based on network structure and often fail to address dynamic aspects of flow and time which are crucial when evaluating the real behaviour of the system and the related risks. To study these dynamics, we approximate a nonlinear high dimensional system with a series of linear operators and accordingly employ time series of systemic risk measures. These can be only calculated after completion of perturbation analysis for all stations.

*Systemic Influence* and *Systemic Fragility*, Fig 6, are systemic risk measures that were first introduced in [13] and further developed in [12]. Systemic influence estimates the repercussions of a disruption at a certain station to the rest of the network when the intensity of the repercussion exceeds a certain threshold. Systemic influence does not necessarily correlate to known centrality measures as it also reveals influential paths. Systemic fragility estimates the vulnerability or exposure of a station to disruptions that occur elsewhere in the network.

For a given threshold $\gamma$, we define the absolute impact of a disrupted node $i$ on some node

$$j \neq i \text{ as } W_{ij} = \begin{cases} |\tilde{\pi}_j - \pi_j|, & \text{if } \frac{|\tilde{\pi}_j - \pi_j|}{\pi_j} > \gamma \\ 0, & \text{otherwise} \end{cases} \text{ and the systemic influence as } I_i = \frac{\sum_j W_{ij}}{\max \sum_j W_{:j}}. \text{ Sys-}$$

temic fragility is then defined as $\phi_i = \frac{\sum_j F_{ji}}{\max \sum_j F_{j:}}$, where $F_{ji} = \begin{cases} 1, & \text{if } W_{ji} > 0 \\ 0, & \text{otherwise} \end{cases}$.

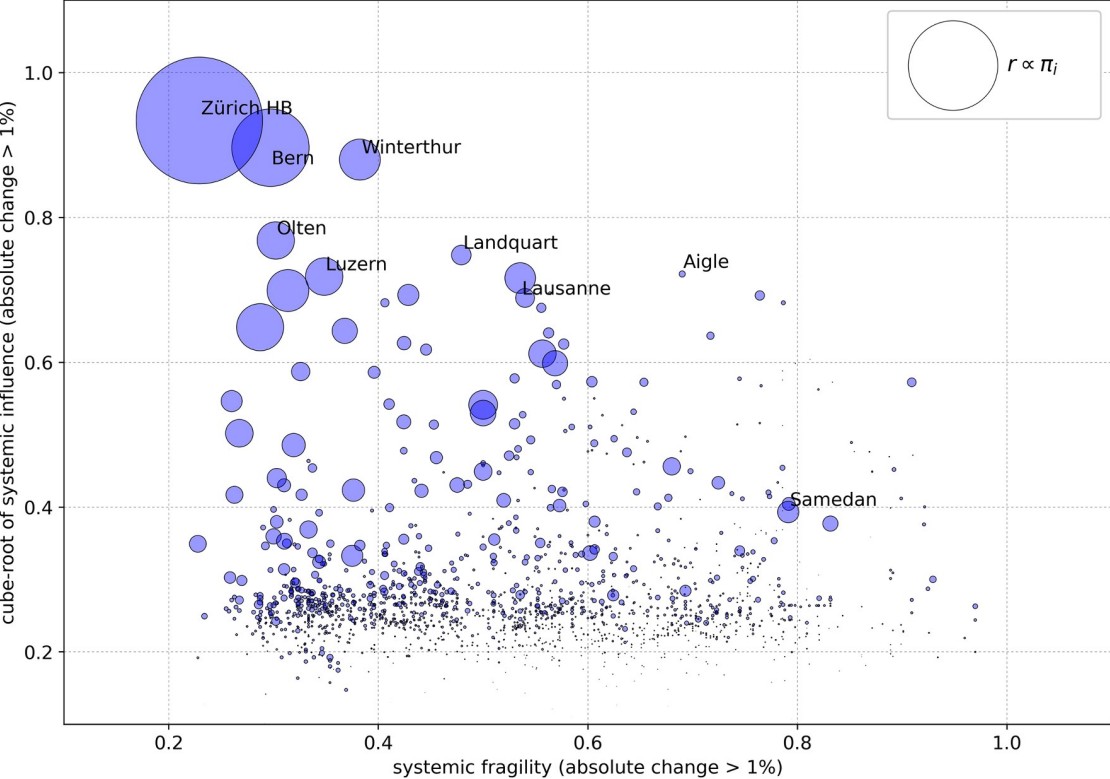

**Fig 6. Systemic influence and systemic fragility of all Swiss train stations.** Median values over a period of one month. Big and influential stations tend to be also less fragile, but this is not a universal trend.

The central hypothesis behind systemic influence is that certain stations with relatively low $\pi_i$ might disproportionately affect the network when disrupted and are therefore very influential. Descriptive statistics also show that this measure, in contrast to centrality measures, can reveal influential groups of stations along certain paths, as we will see in the next section. The intuition behind systemic fragility is that stations in well connected neighborhoods can better absorb shocks than stations that are highly dependent on just a few other nodes and are therefore more fragile.

## Dynamics of the system in time

Best and worst case assessments, community detection and identification of critical stations can be facilitated with the use of descriptive statistics. Simple indicators, such as minimum, maximum, median and standard deviation, efficiently summarize the trends at an aggregate level and highlight areas that require closer attention, both important criteria for experts and decision makers alike.

We identified seven measures that best capture the temporal dynamics of the system: inflow, outflow, stationary distribution, weakly connected sub-communities, remoteness, systemic influence and systemic fragility. Inflow and outflow, which are the respective frequencies of inbound and outbound trains for each node, were found to be practically identical for the Swiss railway network. Because this might not be the case for other systems, we do not consider the measures as redundant, Fig 7.

According to Fig 7, the high variation of flows in the west part of the country caused shifts in the first and second eigenvector, as expressed by the stationary distribution and by the weakly connected sub-communities. This variation might be related to the Swiss autumn holidays. As previously mentioned, systemic influence displays interesting path-like patterns. On the other hand, the patterns ensuing from systemic fragility are remarkably continuous.

Of particular interest is the area around Olten, see S2 Fig, which is known to host the *null-point* stone of the Swiss railway network. The area displays high resistance to perturbations even at maximum fragility. There is historical evidence [33] that this part of the network belonged to the Swiss Central Railways (SCB), a 19[th] century company that was founded with the primary goal to construct a railway cross with its center at Olten. This is a key finding that suggests a link between systemic fragility and network growth. In that sense, there is a historical backbone of low fragility rooted in Zurich, Bern and Olten. We postulate that systemic fragility is a kind of network growth rings, a claim which needs to be verified also for other networks in further research.

## Rankings

We conclude the presentation of our results with some rankings based on the special measures introduced here: remoteness, influence and fragility. The complementary character of the selected measures becomes clear when we consider the top ten stations of the respective high and low segments, Fig 8. Each of the metrics captures unique qualities of the system.

Table 1 lists the top 10 most remote stations. The common characteristic of these stations is that they are extremely isolated. The first four are consecutive stops of the second highest rail crossing in Europe.

Table 2 lists the top 10 most influential stations. All of them are situated at important crossroads. Our assumption regarding the existence of very influential stations with low $\pi_i$ is confirmed by Aigle and S. Antonino.

Table 3 lists the top 10 stations with highest systemic fragility. They are all stops of the highest rail crossing in Europe. This route connects Switzerland to Italy.

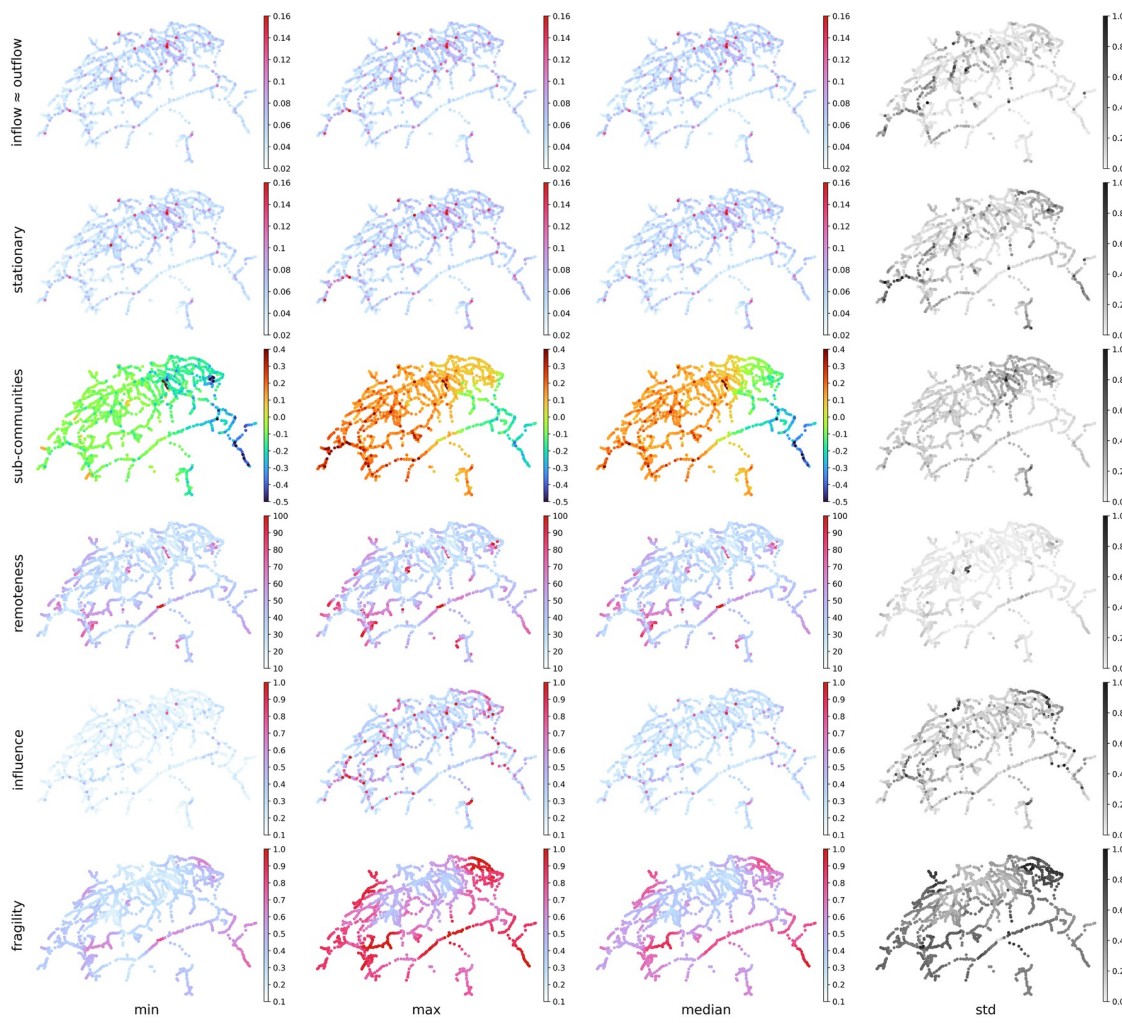

**Fig 7. System overview over a period of one month.**

## Conclusion

StationRank provides a comprehensive aggregate methodology for spatiotemporal analysis and evaluation of actual railway dynamics. The respective data is updated on a daily basis, so given the simplicity and efficiency of the algorithm, it is very easy to achieve near real-time

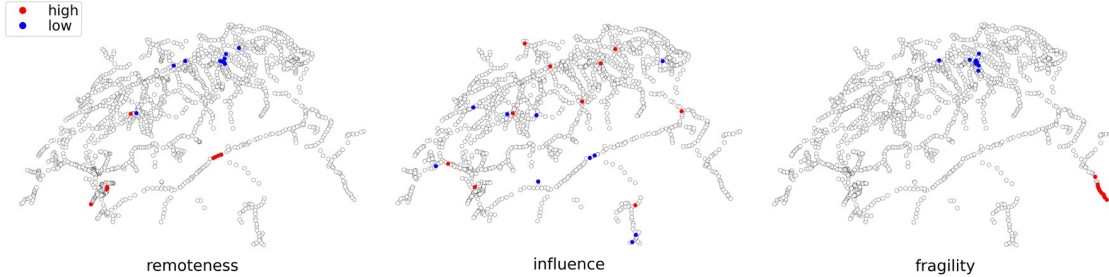

**Fig 8. Ten highest and lowest values of selected measures, 1 month median.**

**Table 1. Top ten most remote stations, 1 month median.**

| Rank | Name | $\pi_i$ | remoteness | influence | fragility |
|---|---|---|---|---|---|
| 1 | Realp DFB | 0.011913 | **145.616835** | 0.106894 | 0.794643 |
| 2 | Tiefenbach DFB | 0.015009 | **135.624551** | 0.097143 | 0.776786 |
| 3 | Furka DFB | 0.045846 | **123.926770** | 0.199714 | 0.794643 |
| 4 | Muttbach-Belvédère | 0.015009 | **109.513526** | 0.086690 | 0.803571 |
| 5 | Col-de-Bretaye | 0.024849 | **98.983666** | 0.168513 | 0.705882 |
| 6 | Bouquetins | 0.031308 | **96.689770** | 0.207581 | 0.712871 |
| 7 | Niederbottigen BN | 0.010464 | **96.215162** | 0.000000 | 0.448276 |
| 8 | Villars-sur-Ollon Golf | 0.031308 | **94.288154** | 0.207223 | 0.715686 |
| 9 | Col-de-Soud | 0.031308 | **91.614750** | 0.207225 | 0.712766 |
| 10 | Champéry | 0.030092 | **90.336117** | 0.183455 | 0.714286 |

**Table 2. Top ten most influential stations, 1 month median.**

| Rank | Name | $\pi_i$ | remoteness | influence | fragility |
|---|---|---|---|---|---|
| 1 | Zürich HB | 0.201419 | 14.056603 | **0.933924** | 0.229167 |
| 2 | Bern | 0.170952 | 15.174429 | **0.896393** | 0.297030 |
| 3 | Winterthur | 0.138236 | 17.549854 | **0.879965** | 0.382353 |
| 4 | Olten | 0.134083 | 15.276948 | **0.768222** | 0.302083 |
| 5 | Landquart | 0.108044 | 26.432022 | **0.748132** | 0.479167 |
| 6 | Aigle | 0.073643 | 30.187482 | **0.721957** | 0.690000 |
| 7 | Luzern | 0.134281 | 19.929157 | **0.718386** | 0.348315 |
| 8 | Lausanne | 0.125590 | 23.792595 | **0.716358** | 0.535354 |
| 9 | Basel SBB | 0.139321 | 18.930696 | **0.699123** | 0.313725 |
| 10 | S. Antonino | 0.057358 | 36.059869 | **0.695352** | 0.563758 |

results. Depending on the scope of application, the results can be used in the form of interactive perturbation analyses, holistic spatiotemporal evaluations of the system's behavior and of course various rankings, thus enabling valuable insights. In contrast to disaggregated approaches such as agent-based models, aggregated models are helpful for validation of existing networks, not for designing networks from scratch.

**Table 3. Top ten most fragile stations, 1 month median.**

| Rank | Name | $\pi_i$ | remoteness | influence | fragility |
|---|---|---|---|---|---|
| 1 | Brusio | 0.055629 | 61.064387 | 0.244052 | **0.969697** |
| 2 | Campocologno | 0.067788 | 61.941892 | 0.263069 | **0.969697** |
| 3 | Campascio | 0.045512 | 62.149911 | 0.200012 | **0.969697** |
| 4 | Miralago | 0.047410 | 60.130656 | 0.219155 | **0.959596** |
| 5 | Tirano | 0.041149 | 62.288009 | 0.192476 | **0.933962** |
| 6 | Poschiavo | 0.075144 | 57.133147 | 0.300235 | **0.929293** |
| 7 | Cadera | 0.042235 | 57.280377 | 0.226393 | **0.925532** |
| 8 | Le Prese | 0.061948 | 58.911337 | 0.287173 | **0.924528** |
| 9 | Li Curt | 0.052842 | 58.446555 | 0.400303 | **0.921348** |
| 10 | Ospizio Bernina | 0.059130 | 53.876171 | 0.376105 | **0.920792** |

## Supporting information

**S1 Fig. Sub-network example in the canton of Grisons.**
(TIF)

**S2 Fig. Network overview in topological and geographical context.**
(TIF)

**S3 Fig. Geographic distribution of transpose mean first passage time.**
(TIF)

**S4 Fig. Single day view of the spectral clusters (2$^{nd}$ eigenvector).**
(TIF)

**S5 Fig. Single day view of a disruption at St. Gallen.**
(TIF)

**S6 Fig. Single day view of systemic influence.**
(TIF)

**S7 Fig. Single day view of systemic fragility.**
(TIF)

**S8 Fig. Time series of the stationary distribution (1$^{st}$ eigenvector).**
(TIF)

**S9 Fig. Time series of systemic influence.**
(TIF)

**S10 Fig. Time series of systemic fragility.**
(TIF)

## Acknowledgments

The corresponding author would like to thank Mirjam Petri for the preliminary discussions on the topic. The authors would like to thank the Swiss Federal Railways (SBB) for maintaining the data and for their valuable comments on the manuscript.

## Author Contributions

**Conceptualization:** Georg Anagnostopoulos, Vahid Moosavi.

**Data curation:** Georg Anagnostopoulos.

**Formal analysis:** Georg Anagnostopoulos, Vahid Moosavi.

**Investigation:** Georg Anagnostopoulos, Vahid Moosavi.

**Methodology:** Vahid Moosavi.

**Project administration:** Georg Anagnostopoulos.

**Software:** Georg Anagnostopoulos.

**Supervision:** Vahid Moosavi.

**Validation:** Vahid Moosavi.

**Visualization:** Georg Anagnostopoulos.

**Writing – original draft:** Georg Anagnostopoulos, Vahid Moosavi.

**Writing – review & editing:** Georg Anagnostopoulos.

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
