## [Decision Letter · Decision Letter 0]

17 Aug 2020

PONE-D-20-17563

StationRank: Aggregate dynamics of the Swiss railway

PLOS ONE

Dear Dr. Anagnostopoulos,

Thank you for submitting your manuscript to PLOS ONE. After careful consideration, we feel that it has merit but does not fully meet PLOS ONE’s publication criteria as it currently stands. Therefore, we invite you to submit a revised version of the manuscript that addresses the points raised during the review process.

Please take careful note of the points indicated by each of the referees as both thought the paper required a number of changes before it could be accepted, although these points/changes varied to some degree from referee to referee.

We look forward to receiving your revised manuscript.

Kind regards,

Ben Webb, Ph.D.

Academic Editor

PLOS ONE

2. Our internal editors have looked over your manuscript and determined that it is within the scope of our Cities as Complex Systems Call for Papers. This collection of papers is headed by a team of Guest Editors for PLOS ONE: Marta Gonzalez (University of California, Berkeley) and Diego Rybski (Potsdam Institute for Climate Impact Research).

The Collection will encompass a diverse and interdisciplinary set of research articles applying the principles of complex systems and networks to problems in urban science.  Additional information can be found on our announcement page: https://collections.plos.org/s/cities.

If you would like your manuscript to be considered for this collection, please let us know in your cover letter and we will ensure that your paper is treated as if you were responding to this call. If you would prefer to remove your manuscript from collection consideration, please specify this in the cover letter.

Reviewers' comments:

Reviewer's Responses to Questions

**Comments to the Author**

1. Is the manuscript technically sound, and do the data support the conclusions?

Reviewer #1: Yes

Reviewer #2: Yes

2. Has the statistical analysis been performed appropriately and rigorously? 

Reviewer #1: No

Reviewer #2: Yes

3. Have the authors made all data underlying the findings in their manuscript fully available?

Reviewer #1: Yes

Reviewer #2: Yes

4. Is the manuscript presented in an intelligible fashion and written in standard English?

Reviewer #1: Yes

Reviewer #2: Yes

5. Review Comments to the Author

Reviewer #1: The manuscript "StationRank:Aggregate dynamics of the Swiss railway" by Georg Anagnostopoulos and Vahid Moosavi presents a Markov Chain (MC) framework to analyse daily aggregated itineraries of the swiss railway systems. They use their MC framework to asses the congestion, resilience and fragility of the railway network.

I think that this framework is interesting. It is simple and does not require much computation power and allows one to draw interesting insights from the data. However, this manuscript is not fit for publication in its current state. My main issues are the following:

- The authors do not place this study in context. There are many existing tools for the analysis of public transportation that can be much more detailed (using agent-based model for example, as in [1]).

In the introduction, the authors should explain what it is they want to do, summarize previous works, explain why their approach is relevant and what is their contribution. In the conclusion, they could elaborate on the limitations of their approach and on possible improvements.

- The methodology is not well explained. Many points are unclear. For example, it is not clear how the transition matrices are computed. The authors make many assumptions in their model but do not explain or justify them.

There are also many smaller issues that I list below::

- eq. 1, define S and x

- p. 2: How is P_ij computed? How is the daily data aggregated?

- p. 2 :Why do you use a 1min discretization? why not 10min or each km for example?

- p. 2: define L-space and P-space.

- p. 2: Describe the final networks of stop and transits. How many nodes are stations? You could show a sub-network as an example.

- p. 3: Explain better Fig. 2. what are the labels? what are the nodes? This is really not clear.

- p. 3: have you tried different values of alpha in the teleportation trick? You could discuss this a little bit more. Certainly passengers do not teleport in real life. What kind of artefacts does this create? We could also imagine other approaches to make the network strongly connected. For example, nodes connecting to outside of Switzerland could be all connected together.

- p. 4: You should also discuss the fact that all the quantities you uses are evaluated at stationarity, but the system is actually never stationary. What does the stationary distribution represents in this case?

- p. 4: About the Kemeny constant, the authors write that it can be described as the "average expected time (steps) from any given state (origin) to any other random state (destination)". This is interesting, but surely, the "expected time from any given state to any other random state" has a very heterogeneous distribution. It would be interesting to also have a measure of the standard deviation of this distribution.

- p. 4: "For the Swiss railway network, we calculated that K ≈ 30.5, ±1 minute." How is the error (and average) calculated?

- p. 5: "Linear reduction of a station’s activity by multiple perturbations of the transition

probability matrix is more suitable for a real-world network than simple node removal." Could you explain why?

- Fig. 3: Show the dates for each plot and do not use a diverging colourmap for a continuously increasing quantity. The position of the white colour is arbitrary. (see https://matplotlib.org/3.1.0/tutorials/colors/colormaps.html)

- p. 5: "In the first weekend, high probabilities were redistributed in the west part of the country. Furthermore, some of the lowest probabilities were reduced near the end of the month."

This is not clear at all on Fig. 3. You could show of the variation of pi in the East and West.

- p. 5: "namely the fact that the first weekend of October developed a completely distinct

dynamic which might be related to the beginning of the Swiss autumn holidays." We see a clear shift, and this is quite nice, but I would not say that it is "a completely distinct dynamic".

It would be nice if you could assess if it is related to the beginning of the Swiss autumn holidays by looking at some other statistics in the data that does not rely on the MC framework.

- p. 6: "Known network measures, such as various centrality measures [8, 13, 21], are purely

based on network structure and often fail to address dynamic aspects of flow and time

which are crucial when evaluating the real behaviour of the system and the related risks."

This is not true, most centrality measures are based on an underlying diffusion process. E.g. degree centrality can be seen as the ranking given by the stationary distribution of a random walk. Same of page rank. Moreover, here you assess systems at stationarity, so you are not rely considering the dynamic of the system apart from daily evolution of the stationarity.

- p. 6: Explain the Bruess Paradox.

- p. 7: Explain the intuition behind the definition of systemic influence and systemic fragility.

- p. 7: "This is a striking finding that suggests a link between systemic fragility and network growth, a claim which needs to be verified also for other networks in further research." Interesting, but there is no need to use bold.

- Fig. 6: Give the names of a few stations.

- Fig. 7. Do not use divergent colourmaps for influence and fragility.

- Fig. 8. How many highest and lowest values are you showing?

[1] P Manser, H Becker, S Hörl, KW Axhausen, Designing a large-scale public transport network using agent-based microsimulation, Transportation Research Part A: Policy and Practice, 2020

Reviewer #2: The study investigates the application of Markov chain (MC) theory to the analysis of the Swiss railroad network historical data. The analytical approach takes inspiration from Goolge’s PageRank algorithm (Page et al., 1999) and follows closely the mathematical and analytical framework established by Crisostomi et al. (2011) for MC applications to transportation networks. The study uses open access data covering a moth worth of activities for the Swiss railroad network to analyze its aggregate dynamics and sensitivity to perturbation in flow.

While the article introduces virtually no new theoretical concepts to the problem, it does present an interesting adaptation of the analytical approach to actual historical data for the Swiss railroad network. The article is well written and easy to follow, however the theoretical treatment of some of the concepts utilized in the study are glossed over rather quickly. I suggest the authors take the following comments/suggestions into consideration to address these shortcomings; please see attachment.

6. PLOS authors have the option to publish the peer review history of their article (what does this mean?). If published, this will include your full peer review and any attached files.

Reviewer #1: No

Reviewer #2: **Yes: **Sinan Salman, PhD

---

## [Author Response · Author response to Decision Letter 0]

21 Oct 2020

Responses to editor’s comments: 

https://journals.plos.org/plosone/s/file?id=wjVg/

PLOSOne_formatting_sample_main_body.pdf and

Response:

Figure names have been changed according to PLOS ONE’S style requirements.

Author affiliations have also been reformatted.

2. Our internal editors have looked over your manuscript and determined that it is within the scope of our Cities as Complex Systems Call for Papers. This collection of papers is headed by a team of Guest Editors for PLOS ONE: Marta Gonzalez (University of California, Berkeley) and Diego Rybski (Potsdam Institute for Climate Impact Research).

The Collection will encompass a diverse and interdisciplinary set of research articles applying the principles of complex systems and networks to problems in urban science. Additional information can be found on our announcement page: https://collections.plos.org/s/cities.

If you would like your manuscript to be considered for this collection, please let us know in your cover letter and we will ensure that your paper is treated as if you were responding to this call. If you would prefer to remove your manuscript from collection consideration, please specify this in the cover letter.

Response:

We also feel that PLOS ONE’s Cities as Complex Systems Call for Papers is a great fit for our manuscript. Therefore, we would like our paper to be considered for this collection.

Responses to reviewer #1:

The manuscript "StationRank: Aggregate dynamics of the Swiss railway" by Georg Anagnostopoulos and Vahid Moosavi presents a Markov Chain (MC) framework to analyze daily aggregated itineraries of the Swiss railway systems. They use their MC framework to asses the congestion, resilience and fragility of the railway network.

I think that this framework is interesting. It is simple and does not require much computation power and allows one to draw interesting insights from the data. However, this manuscript is not fit for publication in its current state. My main issues are the following:

- The authors do not place this study in context. There are many existing tools for the analysis of public transportation that can be much more detailed (using agent-based model for example, as in Manser et al. 2020). In the introduction, the authors should explain what it is they want to do, summarize previous works, explain why their approach is relevant and what is their contribution. In the conclusion, they could elaborate on the limitations of their approach and on possible improvements.

Response:

The present study provides a stochastic framework for studying interactions between localities (stations) that are connected via railway infrastructure. At the center of our analysis is the station,

not the train. We construct rankings of stations by means of aggregating daily train flows. In terms of scope, we cover the whole Swiss railway network over a period of one month. Our hypothesis is that we can obtain a quite detailed idea of the spatiotemporal dynamics involved just by utilizing a single openly accessible data set, at low computational cost and without assumptions involving substantial domain-specific knowledge.

Existing parametric tools, such as agent-based models, have been applied to many engineering problems related to complex systems, including analysis of public transportation. Nevertheless, their success relies on the acceptance of a predefined set of properties (Moosavi 2017) whose gradual increase in pursuit of realism is bound by the curse of dimensionality (Bellman 2015). 

On the other hand, purely stochastic models implicitly capture complexity in a probabilistic manner. In PageRank (Page et al. 1999) for example, websites are indirectly represented by their relations to other websites without relying on predefined semantics. Since then, there have been multiple adaptations of this idea, best summarized in “PageRank beyond the web” (Gleich 2015).

Our approach builds on the foundations laid out by Crisostomi et al. (2011) on road network dynamics and adapts it for a real railway network by introducing a novel model architecture. StationRank preserves actual train stations by avoiding aggregation in space and is thus much more detailed than earlier (Moosavi and Hovestadt 2013) or current (Hackl and Adey 2019) approaches that employ spatial binning. Finally, this study complements previous research by Moosavi and Isacchini (2017) on systemic risk (Battiston et al. 2012) with spatial aspects.

In contrast to disaggregated approaches such as agent-based models, aggregated models are helpful for validation of existing networks, not for designing networks from scratch.

- The methodology is not well explained. Many points are unclear. For example, it is not clear how the transition matrices are computed. The authors make many assumptions in their model but do not explain or justify them.

Response:

The majority of existing stochastic traffic models focus on road networks and have established methods for calculating the respective transition matrices. For example Crisostomi et al. (2011) or Salman and Alaswad (2018) consider turning probabilities at road intersections and mean travel times at road segments in order to construct the transition matrix. Moosavi and Hovestadt (2013) sample GPS traces of taxicabs with a rectangular grid and construct the Markov matrix based on grid cell transition times. Other approaches, as in Hackl and Adey (2019), solve traffic assignment optimization problems based on synthetic behavioral data and domain-specific assumptions which result in origin-destination (OD) matrices.

Our methodology for computing the transition probability matrices is more tailored towards utilization of actual public transport itineraries (trips). Trips are stop sequences with continuous departure and/or arrival times at discrete locations (stations). Every trip consists of dwell and running times (Şahin 2017). After discretization (Doytchinov and Irby 2010), dwell times are assigned to actual stations (dwell states) and running times are assigned to virtual transit stations (running states). We can think of these stations (actual or virtual) as Markovian states . By simply counting the total number of transitions from one state to another state , we obtain a weight matrix A. Off-diagonal entries Aij represent change of state and diagonal entries Aii represent preservation of state. We can show that it is pretty straightforward to obtain the transition probabilities Pij given Aij.

There are also many smaller issues that I list below:

- eq. 1, define S and x

Response:

 denotes the position at time step of a particle following a random walk. According to Crisostomi et al. (2011) “vehicles do not actually follow a random walk; however, by analogy with similar approaches like the PageRank model … , the underlying stochastic process is viewed as a mechanism for obtaining useful information, rather than a literal representation of the behaviour of a single vehicle”.

- p. 2: How is P_ij computed? How is the daily data aggregated?

Response:

Regarding the aggregation, we group stops by trip id and day of operation, a property that resides in the actual data. A day of operation is not the same as a calendar day. Consider the following: a single trip around midnight might span two calendar days, but occurs in the same day of operation.

Then, for each day of operation, we convert all trips in the form of special continuous trajectory objects (Graser and Dragaschnig, 2020) which we descretize: http://anagno.com/trajectory.html. The resulting discrete trajectories are in fact Markovian state sequences.

Example: (A), (A -> B), (A -> B), (B), (B), (B -> C), ... , (X -> Y), (Y), (Y) , (Y -> Z), (Z)

Letters denote actual train stations. By computing the daily total number of observed transitions between a state Si = (A) and a state Sj = (A -> B), we get Aij.

The weight matrix A is actually the adjacency matrix of a directed multi-graph D (with self-loops). From D, we extract the strongly connected component G , using an algorithm by Pearce (2005). Finally, we compute the row-stochastic transition probability matrix P from the adjacency matrix of G by ensuring that every row sums up to 1.

- p. 2: Why do you use a 1min discretization? why not 10min or each km for example?

Response: 

At every time step k, the random variable Xk should be unambiguously assigned to just a single state Sk. It doesn't matter if Sk is an actual train station (dwell state) or a running state between stations, as long as the function k -> Sk is injective (one to one). Now, if we consider 10min discretization, a suburban train with frequent stops might have traversed multiple states within this time step, which leads to a contradiction (state ambiguity). For other systems, such as bus or tram networks, one might even want to consider a discretization of 30sec. Of course there exist other valid Markovian frameworks with 10min time step (Moosavi and Hovestadt 2013), but they employ spatial binning which is not meaningful in our setup.

- p. 2: define L-space and P-space.

Response:

In the context of public transport networks (PTN), a space is a way to define network topology. Some examples of spaces are L-space, P-space, B-space and C-space (von Ferber et al. 2009). Each station in L-space is represented by a node. Links between nodes indicate that the respective stations are consecutively served by at least one route (Berche et al. 2010; von Ferber et al. 2009). If stations are connected only when they are consecutive stops of at least one vehicle, then we have a space-of-stops as a special case of L-space. If stations are connected only when there is a physical direct link (such as rail tracks) with no other station in between, then L-space is actually a space-of-stations (Kurant and Thiran 2006). Space-of-stops coincide with space-of-stations when vehicles stop at every station. An extended L'-space allows multiple weighted links between nodes.

In P-space any two stations, consecutive or not, that share at least one common route are linked. This representation is also called space-of-changes (Kurant and Thiran 2006) because each route results in a clique including all stations that can be reached without changing vehicle. B-space is a bipartite graph where stations are linked to serving routes. Finally, C-space does not include any stations at all, only routes. In C-space connected routes share at least one station.

Our network is defined in L'-space. We introduce a generic space-of-states, where a state can be a stop or a transit. Space-of-states explicitly captures temporal dynamics as it does not assume any fixed topology (e.g. roads or rail-tracks) and can easily support construction of discrete-time, finite-state homogeneous MCs without spatial sampling. This novel approach is easily applicable also to other systems, such as air traffic.

- p. 2: Describe the final networks of stop and transits. How many nodes are stations? You could show a sub-network as an example.

Response:

In total, 1660 nodes are stations. Our network incorporates temporal as well as modal aspects. Dwell states (actual stations) might be directly or indirectly connected via running states (transits). An example sub-network in the area of Grisons can be found in the supporting information section.

- p. 3: Explain better Fig. 2. what are the labels? what are the nodes? This is really not clear.

Response:

The nodes are Markovian states. We distinguish between dwell states which are actual stations, such as (A), and running states which are transits between stations, such as (A -> B). Some times stations like (B) and (C) might be both directly and indirectly linked due to differences in speed of connecting trains. A disruption at node (C) can be caused by means of reducing all of its inflows.

- p. 3: have you tried different values of alpha in the teleportation trick? You could discuss this a little bit more. Certainly passengers do not teleport in real life. What kind of artifacts does this create? We could also imagine other approaches to make the network strongly connected. For example, nodes connecting to outside of Switzerland could be all connected together.

Response:

This is absolutely true. Teleportation might distort network performance metrics such as mean first passage times and Kemeny’s constant. It also adversely affects sparsity of the transition probability matrices which means longer computation times. Simple heuristics fail as there might be whole groups of stations that behave as either absorbing or null-recurrent. A nice overview of all possible network artifacts, including in-components, out-components, in-tendrils, out-tendrils, tubes and disconnected components is provided by Broder et al. (2000). While teleportation works for web pages, it is less meaningful in the context of transportation systems, so we limited our analysis to the strongly connected component which can be found by efficient depth-first search (Pearce 2005).

- p. 4: You should also discuss the fact that all the quantities you use are evaluated at stationarity, but the system is actually never stationary. What does the stationary distribution represents in this case?

Response:

The term stationary in the context of Markov Chains is not referring to a stationary system. For an example of a stationary model, see Dorbritz (2012). Here the stationary distribution π represents occupancy of stations at a theoretical state of equilibrium, it shows a trend. Generally, π is just one property of the underlying Markovian system as described by the transition probability matrix P. The matrix P in turn can support much more demanding analysis such as spectral clustering, calculation of mean first passage times, network resistance estimation, link recommendation, etc.

- p. 4: About the Kemeny constant, the authors write that it can be described as the "average expected time (steps) from any given state (origin) to any other random state (destination)". This is interesting, but surely, the "expected time from any given state to any other random state" has a very heterogeneous distribution. It would be interesting to also have a measure of the standard deviation of this distribution.

Response:

Thank you for commenting on this. We decided to add a new subsection in order to satisfactorily cover this point: lines 185-207.

- p. 4: "For the Swiss railway network, we calculated that K ≈ 30.5, ±1 minute." How is the error (and average) calculated?

Response:

We calculate time series of K, with the two methods previously described, and we normalize them by the respective aggregation times. The reported result was true for a smaller network with fixed aggregation times of 1440min (24 hours). Now we consider a total of 1660 stations and aggregate by day of operation rather than calendar day, so times also vary. The new result is quite interesting in that it shows a hard upper boundary of 41min. In a way, bigger networks are “slower”.

- p. 5: "Linear reduction of a station’s activity by multiple perturbations of the transition probability matrix is more suitable for a real-world network than simple node removal."

Could you explain why?

Response:

Perturbation of the transition probability matrix enables link capacity reduction (Dobritz 2012). In many real-world scenarios only certain connections of a station fail or switch to reduced operation.

- Fig. 3: Show the dates for each plot and do not use a diverging colourmap for a continuously increasing quantity. The position of the white colour is arbitrary.

(see https://matplotlib.org/3.1.0/tutorials/colors/colormaps.html)

Response:

In Fig 4 and Fig 5 we show the dates. Regarding the colors, we have the following mapping:

 • remoteness, inflow, outflow, stationary distribution, fragility, influence→ CET-L191

 • standard deviation → CET-L2

 • spectral clustering → turbo

 • perturbations → seismic

- p. 5: "In the first weekend, high probabilities were redistributed in the west part of the country. Furthermore, some of the lowest probabilities were reduced near the end of the month."

This is not clear at all on Fig. 3. You could show of the variation of pi in the East and West.

Response:

We replaced the time series in Fig 3 with a single day view. The variation of is shown in Fig 7.

- p. 5: "namely the fact that the first weekend of October developed a completely distinct dynamic which might be related to the beginning of the Swiss autumn holidays." We see a clear shift, and this is quite nice, but I would not say that it is "a completely distinct dynamic". It would be nice if you could assess if it is related to the beginning of the Swiss autumn holidays by looking at some other statistics in the data that does not rely on the MC framework.

Response:

The distinct dynamic characterization is omitted in the revised manuscript. The patterns initially observed, persist also in the current version that analyzes considerably more data. The observation is consistent among multiple measures, including non Markovian ones such as inflow and outflow. This is clear in Fig 7 and can be easily verified with our app: https://stationrank.herokuapp.com/.

- p. 6: "Known network measures, such as various centrality measures [8, 13, 21], are purely based on network structure and often fail to address dynamic aspects of flow and time which are crucial when evaluating the real behavior of the system and the related risks."

This is not true, most centrality measures are based on an underlying diffusion process. E.g. degree centrality can be seen as the ranking given by the stationary distribution of a random walk. Same of page rank. Moreover, here you assess systems at stationarity, so you are not really considering the dynamic of the system apart from daily evolution of the stationarity.

Response:

Certainly the stationary distribution can be viewed as a kind of centrality measure. Here the terminology might seem counter-intuitive, but stationarity is actually just a property of a compact dynamical system representation, in this case of a Markov Chain (Froyland 2001). We do not actually consider the evolution of stationarity, but rather approximate a non-linear dynamical system with a series of linear operators.

- p. 6: Explain the Braess Paradox.

Response:

In the current version we place more emphasis on better explaining the inner workings of the Kemeny constant itself, thus omitting any references to ΔΚ or the Braess Paradox.

- p. 7: Explain the intuition behind the definition of systemic influence and systemic fragility.

Response:

Systemic influence measures the repercussions of a disruption at a certain station to the rest of the network when the intensity of the repercussion exceeds a certain threshold. Systemic influence does not necessarily correlate to known centrality measures as it also reveals influential paths, Fig 7. Systemic fragility measures the vulnerability or exposure of a station to disruptions that occur elsewhere in the network. 

- p. 7: "This is a striking finding that suggests a link between systemic fragility and network growth, a claim which needs to be verified also for other networks in further research." Interesting, but there is no need to use bold.

Response:

Indeed, bold is not needed and has been removed.

- Fig. 6: Give the names of a few stations.

Response:

We added the names and also some provision to avoid overlaps: https://stationrank.herokuapp.com/.

- Fig. 7. Do not use divergent colourmaps for influence and fragility.

Response:

See previous response with regards to Fig. 3

- Fig. 8. How many highest and lowest values are you showing?

Response:

We show the 10 highest and 10 lowest values in alignment with the corresponding tables.

Responses to reviewer #2:

The study investigates the application of Markov chain (MC) theory to the analysis of the Swiss railroad network historical data. The analytical approach takes inspiration from Goolge’s PageRank algorithm (Page et al., 1999) and follows closely the mathematical and analytical framework established by Crisostomi et al. (2011) for MC applications to transportation networks. The study uses open access data covering a month worth of activities for the Swiss railroad network to analyze its aggregate dynamics and sensitivity to perturbation in flow.

While the article introduces virtually no new theoretical concepts to the problem, it does present an interesting adaptation of the analytical approach to actual historical data for the Swiss railroad network. The article is well written and easy to follow, however the theoretical treatment of some of the concepts utilized in the study are glossed over rather quickly. I suggest the authors take the following comments/suggestions into consideration to address these shortcomings:

General comments:

1. While the study focuses on railway traffic, the analysis, as I understand it, was based on trains data. As such the data does not actually represent passengers or goods flow on the network, only trains. While the authors make no claims that their analysis considers actual passengers’ or goods traffic, this should be noted in the data section to help ground the reader on what the analysis actually considers. To understand why this clarification is important, consider the following: a perturbation of say +10% passenger demand homogeneously added to the network can be addressed by many means, one of which could be to increasing trains capacity homogeneously via additional cars; and thus the network analysis, as I believe, will show no impact from the additional flow as the number of trains and their schedules remains constant.

Response:

We absolutely agree. This important clarification is now part of our data section (line 104-105). In future research, it would be interesting to quantify the flows for example in financial terms by combining different data. Still with the detail level available in the data, we show that it is possible to gain interesting insights about the system.

2. The authors do not provide a summary of alternative literature-based approaches used to analyze this problem. Are there other approaches to compare the reported results to?

Response:

There is a sparsity of publications on the spatiotemporal dynamics of the Swiss railway system (or other systems in that regard) as literature is dominated by studies focusing on road traffic. For example an early paper by Erath et al. (2009) on a “graph-theoretical analysis of the Swiss road and railway networks over time”, actually only covers the Swiss freeways, not railways. Later, Dorbritz (2012) developed a methodology for assessing the structural and operational robustness of railway networks with examples from Switzerland, but his model is stationary and acknowledges that it takes “much effort for turning the stationary model into a dynamic one”. However, this is the closest reference known to the authors in terms of comparing the results. Recently, Büchel and Corman (2019) took an aggregated approach to explain spatial and temporal ramifications of an extremely rare disruptive event in statistical terms, but their methodology has a rather forensic character and does not constitute a framework. There are also other approaches such as Hackl and Adey (2019), who omit railways as well.

3. The provided resolutions for Figures 3, 4, 5, 7, and 8 do not allow the reader to properly analyze the presented data. In the case of this reviewer, I was able to use the corresponding figures in the online notebook, however better resolutions or different visualization approaches should be used.

Response:

For figures 3, 4 and 5 following rule applies: When temporal evolution is less pronounced as in Fig 3, we simply show a single day view and move the respective time series plot in the supporting information section. When there is interesting temporal trends as in Fig 4 and Fig 5, we show the time series plot (in better resolution) and include a selected single day view in the supporting information section. In Fig 7 we have changed colors and we have optimized the layout in order to maximize space for the plots, same in Fig 8 with bigger plots.

4. While I applaud the authors for making their analysis and results available online for the reader, and encouraging the reader to work with the open source code in the cited repository, I believe that the majority of the readership may not have the skillset or the time needed to get familiar with and run through such a lengthy analysis code. May I suggest that for the perturbation analysis, the authors provide interactive plots (using bokeh library or Jupyter widgets, as appropriate) in a separate notebook to help the readers experiment with the model results. This would significantly increase the readership’s access to your analysis and results, beyond your article.

Response:

Thank you for this inspiring comment. Based on your insightful suggestions, we have refactored our source code1 extensively in order to strengthen readability, make it more reproducible and also support appropriate visualization frameworks. We provide interactive plots of our results, including risk plots and perturbation analysis, in a separate notebook: https://bit.ly/2I4uAs3. This notebook can be automatically converted to an app: https://stationrank.herokuapp.com/. 

Specific comments:

5. Line 26: Why is there a need to ensure comparability? Shouldn’t the analysis include the entire network and if a route/station is not used on a given day it will show accordingly in the results? What happens to the discarded stations in the analysis? How does this impact the accuracy of the analysis? What does the 75% coverage figure actually mean?

Response:

Again the authors are grateful for such constructive feedback. As recommended, the analysis now includes the entire network, not just 75% which was dictated by bugs in our initial code. Inactive stations or disconnected components are now accordingly shown in our results as “excluded”. We believe that this approach improves the accuracy of our analysis by reducing outliers and by revealing new details that remained hidden in the first version. See for example new weakly connected sub-communities that were detected by the second eigenvector in Fig 4 or the corresponding single day view which we include in the supporting information section.

6. Line 40: resulting matrices description is ambiguous; is it Nx1440 separate matrices or matrices of the size of Nx1440. What do these matrices include?

Response:

These matrices are no longer in use. Their purpose was to store collections of daily trajectories. Part of the extensive refactoring of our code has been the elimination of such ambiguous data structures and their replacement with more flexible, state of the art trajectory collection objects, as defined in the MovingPandas package (Graser and Dragaschnig, 2020).

7. The use of random teleportation in this application is only mentioned in passing. The authors should explain the physical impact of this significant modification to the structure of the Markov chain; i.e. what does this change equate to in the railroad application? How does this treatment compare to that of Page et al. (Google’s PageRank)? What is the impact on the results accuracy and model representation of the problem? What is the number of null-recurrent and absorbing states compared to the network size?

Response:

Random teleportation blurs out the originally sparse transition probability matrix with a homogeneous dense matrix. This is a relatively high price to pay in order to achieve ergodicity as it significantly alters network topology and results in unnecessarily long computation times. There are also other problems such as exploding mean first passage times and unacceptable values for the Kemeny constant. The cleanest solution in mathematical terms is to refrain from teleportation as in Salman and Alaswad (2018). Because the number of null-recurrent and absorbing stations compared to network size is in the vicinity of 7.5%, we still have a strongly connected component near 92.5% of the original network which is an acceptable trade-off. 

8. Line 107: the authors state that E must be singular. The model only requires all rows to sum to zero. Singular matrices do not all satisfy this condition, as such this statement should be omitted.

Response:

Thank you for this correction. We have omitted the statement as requested.

9. Line 108: what are the non-kth entries values of the ek vector?

Response:

The non-kth entries are equal to zero.

10. Line 108: if the teleportation concept is applied, all Pij will be < 1. Hence the statement “if Pip = 1…” is not needed in this analysis.

Response:

The teleportation concept has been abandoned as it resulted to dense matrices. There are quite a few cases where Pip = 1 in the sparse scenario, so the statement is indispensable.

11. The generalized equation (7) is not clearly discussed. It appears S is used here to indicate a subset of inbound (direct and indirect) states from the total set of states, which is again indicated by S. If so, different symbols must be used to establish the differentiation. If not, why would equation (7) consider all states? The generalization from equation (6) from Crisostomi et at. to (7) should be discussed in clearer terms as it is a new assertion made by the authors.

Response:

Your first interpretation of the equation is correct. In this context indicates a subset of inbound states. We establish the differentiation by replacing with . We now discuss our generalized equation more thoroughly in lines 216-224.

12. Line 118: is the 95% homogeneous reduction applied to all incoming links to a particular station, a subset of stations, or all stations in the model? All at once or in a sequence of separate scenarios?

Response:

Here we discuss different scenarios. The actual perturbation is performed on all incoming links.

13. Perpetuation of 95% is a relative measure; applied to a busy hub will emphasize the impact and applied to a remote station will deemphasize the impact. This fact should be mentioned in the article.

Response:

Yes, absolutely. We integrated this remark in lines 225-227.

14. Figure 3 contains a large amount of information such that it is difficult to discern its implications to the network – especially that the provided resolution is not suitable for such a task. I suggest the authors should limit this figure to a single day and show the network congestion and details. A second alternative could be to add the single day figure prior to this figure, and then discuss the findings from Figure 3 in the text in more detail than what is provide in the paragraph between lines 137-141.

Response:

We have replaced Fig 3 with a single day view according to your suggestion. The findings from Fig 3 can be better understood in tandem with the system overview from Fig 7.

15. A similar comment to that of Figure 3 can be made on Figure 4. Consider a single day view.

Response:

In Fig 4 the temporal evolution is much more interesting than in Fig 3, so we keep the overview in the main article and include a single day view in the appendix.

16. The perturbation analysis methodology (steps to generate the results) should be made clearer to the reader. It is not clear if Figures 6 and 7 flow from the analysis done in Figure 5 with a disruption at the Bern station, or another approach was utilized. It is important to highlight the scenario(s) being shown to enable the reader to understand the data being presented. 

Response:

Yes we agree. Fig 5 is just an example of disruption at one station. In the revised manuscript we took the liberty of replacing Bern with an example that shows clear temporal trends, St. Gallen. If one compares the new Fig 5 with Fig 4, then following observation holds: the disruptive effect is minimal when St. Gallen is part of the small sub-component and gets maximized when St. Gallen shifts to the main sub-component which makes sense in a way. Figures 6 and 7 can only be obtained after the perturbation has been performed for all active stations, not only one.

17. The authors should briefly describe the terms systematic influence and systematic fragility and how it applies to the problem at hand. What do they mean in this problem? It is unclear how Wij can be greater than 1, if it is defined as a measure of difference between stationary distribution values that each can only take values between 0 and 1. This part should be edited to clarify the presented concepts.

Response:

The central hypothesis behind systemic influence is that certain stations with relatively low might disproportionately affect the network when disrupted. Descriptive statistics also show that this measure, in contrast to centrality measures, can reveal influential groups of stations along certain paths, see Fig 7. Systemic fragility measures the vulnerability or exposure of a station to disruptions that occur elsewhere in the network. Stations in densely connected areas can better absorb shocks than stations that are highly dependent on a few other nodes. Regarding the mathematical formulation, we do not actually claim that Wij > 1.

18. Figure (6) displays a negative relationship between systematic influence and fragility; is this significant?

Response:

This trend can be now attributed to insufficient data and/or artifacts related to the application of random teleportation. The updated version shows instead a more balanced system. Nevertheless, it is clear that highly influential stations are usually less fragile.

19. In one of the network figures it would be great if the authors point out the location of Bern and Olten stations to the non-Swiss reader.

Response:

In the supporting information section, we have included a single day figure with the location of Bern, Zürich HB and Olten along with information related to network topology.

20. The link between systematic fragility and network growth is not immediately obvious to this reviewer. Please elaborate in the article.

Response:

While it is not surprising that stations around big agglomerations such as Bern or Zurich have low fragility, the case of Olten shows that low fragility might also be related to the historical development of the network. In that sense there is a historical backbone of low fragility rooted in Zurich, Bern and Olten. At the same time we observe extremities of high fragility that were integrated later. We postulate that systemic fragility is a kind of network growth rings.

---

## [Decision Letter · Decision Letter 1]

7 Dec 2020

StationRank: Aggregate dynamics of the Swiss railway

PONE-D-20-17563R1

Dear Dr. Georg Anagnostopoulos,

We’re pleased to inform you that your manuscript has been judged scientifically suitable for publication and will be formally accepted for publication once it meets all outstanding technical requirements.

Kind regards,

Ben Webb, Ph.D.

Academic Editor

PLOS ONE

There are a few remarks/suggestions the referees have for the authors that should be consider for the final version although no changes are required.

Reviewers' comments:

Reviewer's Responses to Questions

**Comments to the Author**

1. If the authors have adequately addressed your comments raised in a previous round of review and you feel that this manuscript is now acceptable for publication, you may indicate that here to bypass the “Comments to the Author” section, enter your conflict of interest statement in the “Confidential to Editor” section, and submit your "Accept" recommendation.

Reviewer #1: (No Response)

Reviewer #2: All comments have been addressed

2. Is the manuscript technically sound, and do the data support the conclusions?

Reviewer #1: Yes

Reviewer #2: Yes

3. Has the statistical analysis been performed appropriately and rigorously? 

Reviewer #1: Yes

Reviewer #2: Yes

4. Have the authors made all data underlying the findings in their manuscript fully available?

Reviewer #1: Yes

Reviewer #2: Yes

5. Is the manuscript presented in an intelligible fashion and written in standard English?

Reviewer #1: Yes

Reviewer #2: Yes

6. Review Comments to the Author

Reviewer #1: I thank the authors for replying to my remarks. They have addressed all of my issues and the manuscript is now much clearer and its results are robust and interesting. I recommend its publication.

I just have a few remarks/suggestions that the authors should consider for the final version.

Namely:

- p.3 " Then, for each day of operation, we convert all trips to special continuous trajectory objects [13] which we descretize: " ext-link-type="uri" xlink:type="simple">https://bit.ly/3iKLx7A." I suggest to add the linked figure in the manuscript (or use Fig. S1). There is also a typo: descretize - discretize

- p.4. You could state the number of dwell states and the number of running states.

In general, the figure captions could have more information so that it is possible the understand a figure just by looking at it.

- Fig. 2 and 3: why use the cube-root of the values? for Fig. 2 the units should be time units, otherwise this figure is rather meaningless.

- Fig. 4 and 5: add colorbars with units to the plots.

- Fig. 5 and S5: what are the units? percentage of change? please clarify it in the figures.

- Fig. 7: clarify the fact that the min, max, median, ... are taken over the temporal dimension. (If I understood).

- Fig. 7: show Olten on one of the panel.

- p. 9: "According to Fig 7, the high variation of flows in the west part of the country caused shifts in the first and second eigenvector, as expressed by the stationary distribution and by the weakly connected sub-communities."

Please describe exactly to which panels of Fig. 7 you refer to support this argument.

- Fig. 8: It could be nice to add the names and ranks of the stations on the plots.

Reviewer #2: All comments have been addressed.

While I understand the authors' statement in the final sentence of the conclusion: "In contrast to disaggregated...". I believe the authors approach can be extended to the redesign of existing networks through evaluation of different extension scenarios, which can be generated manually or algorithmically. I'll leave it to the authors whether they react to this comment or not in their final manuscript.

7. PLOS authors have the option to publish the peer review history of their article (what does this mean?). If published, this will include your full peer review and any attached files.

Reviewer #1: No

Reviewer #2: **Yes: **Sinan Salman

---

## [Editor Report · Acceptance letter]

9 Dec 2020

PONE-D-20-17563R1 

StationRank: Aggregate dynamics of the Swiss railway 

Dear Dr. Anagnostopoulos:

I'm pleased to inform you that your manuscript has been deemed suitable for publication in PLOS ONE. Congratulations! Your manuscript is now with our production department. 

Kind regards, 

on behalf of

Dr. Ben Webb 

Academic Editor

PLOS ONE